# Medicinal Plants Used in Sri Lankan Traditional Medicine for Dengue Fever

Jayani K. Handagala [1], Nishantha Kumarasinghe [2], Charitha L. Goonasekara [3,*] and Anchala I. Kuruppu [2,*]

1    Institute for Combinatorial Advanced Research and Education, General Sir John Kotelawala Defence University, Ratmalana 10390, Sri Lanka
2    Department of Anatomy, Faculty of Medicine, General Sir John Kotelawala Defence University, Ratmalana 10390, Sri Lanka
3    Department of Preclinical Sciences, Faculty of Medicine, General Sir John Kotelawala Defence University, Ratmalana 10390, Sri Lanka
*    Correspondence: charithalg@kdu.ac.lk (C.L.G.); kuruppua@kdu.ac.lk (A.I.K.)

**Abstract:** Dengue fever, a mosquito-borne viral infectious disease caused by the dengue virus, is a significant global health concern, especially in tropical and subtropical regions. Despite preventive efforts, Sri Lanka faces recurring dengue outbreaks, with the Western province being the most affected. Current treatments primarily focus on supportive care, as specific antiviral therapies remain elusive. This review presents an overview of dengue, its clinical presentations, the dengue burden in Sri Lanka, and the potential of Sri Lankan medicinal plants used in traditional medicine for treating dengue. Several plants, such as *Munronia pinnata*, *Azardirachta indica*, *Cissampelos pareira* L., *Carica papaya*, *Zingiber officinale*, *Curcuma longa*, and *Bambusa vulgaris,* show antiviral properties against dengue. The utilization of these medicinal plants in dengue treatment could offer a promising avenue for further research and drug development.

**Keywords:** antiviral activity; dengue; medicinal plants with antiviral activity

## 1. Introduction

### 1.1. Dengue

Dengue fever is a mosquito-borne viral infectious disease that has become a major public health concern internationally. The genome of the dengue virus is a positive single strand of RNA. It belongs to the genus *Flavivirus* of the family *Flaviviridae.* The dengue virus contains a lipid envelop, three structural proteins, and seven nonstructural proteins [1]. Dengue fever is common in humid, warm, tropical climates and is transmitted by mosquitos, mainly by female *Aedes aegypti* and to a lesser extent by female *Aedes albopictus* [2]. These two species of mosquitoes live in tropical, subtropical, and hot climates. They have an aquatic phase and a terrestrial phase in their life cycle. Their eggs, larvae, and pupae belong to the aquatic phase, and the adult mosquitoes belong to the terrestrial phase. Dengue-vector *Aedes* mosquitoes are container breeders, and they lay eggs in a vast variety of natural and artificial wet containers. The water has to be clear and unpolluted. Unless the eggs come into contact with water and hatch, mosquito eggs can remain viable for up to 8 months depending on the environmental conditions [3]. Dengue is believed to be caused by four different serotypes (DENV-1 DENV-2, DENV-3, and DENV-4), which are genetically similar due to sharing 65% of their genome [4].

### 1.2. Clinical Presentation of Dengue

The clinical presentation of dengue infection may vary from asymptomatic to severe illness and can include dengue hemorrhagic fever (DHF) or dengue shock syndrome (DSS). This may even lead to death if not properly managed [5].

Symptomatic infection may present as:

- Undifferentiated febrile illness, in which the absence of clinical features and the diagnosis can only be conducted based on serology or virology.
- Dengue fever, which is considered a mild illness even though massive bleeding may be associated; deaths are rarely reported.
- DHF, in which there is increasing vascular permeability seen even though the febrile-phase clinical presentations are similar to those of dengue fever.
- DSS, in which the clinical presentations are similar to DHF, but the patient develops shock because of severe plasma leakage.
- Unusual dengue, or expanded dengue syndrome, in which patients show severe organ involvement associated with dengue infection and which may be associated with complications of prolonged shock, co-infections, or co-morbidities [6].

The severity of a dengue infection can be higher when people experience a second attack of the infection. Secondary infection is caused by a different serotype of the virus that can pose a significant health risk due to the phenomena of antibody-dependent enhancement (ADE). Contracting the infection with a single serotype confers immunity only to that serotype. When non-neutralizing antibodies developed during the initial infection connect to the virus during a later infection with a different serotype, ADE occurs. Rather than neutralizing the virus, these antibodies allow it to enter into cells, hence increasing the infection and contributing to more severe outcomes, such as dengue DHF or DSS [7–11].

### 1.3. Dengue Burden in Sri Lanka

Dengue fever is endemic in Sri Lanka, and the first serologically confirmed dengue patient was reported in 1962 [12]. In the recent past, Sri Lanka, India, Indonesia, Myanmar, and Thailand rank among the world's 30 most highly endemic countries. In Sri Lanka, dengue infection has a seasonal transmission pattern. It contains two peaks, mostly during monsoon rain periods such as June–July and October–December, with the majority of cases occurring during the June–July monsoon season. At present, dengue cases are reported from almost all districts of Sri Lanka, with the highest case rate reported in the Western Province, which includes the districts of Colombo, Gampaha, and Kaluthara [13]. DHF became endemic in Sri Lanka in 1989, and since then, despite preventive and control activities, the number of dengue fever cases has increased markedly with each epidemic. Since the 1990s, progressively large dengue epidemics have occurred with severe DHF cases. The largest dengue epidemic in Sri Lanka was reported in 2017, with 186,101 cases and 440 deaths, along with a huge economic and social burden. In 2020, 31,162 suspected dengue cases were reported to the Epidemiology Unit of Ministry of Health of Sri Lanka [14]. In 2021, there were 25,067 dengue cases, and in 2022, there were 66,608. In 2023, there had been 66,476 dengue cases reported to the National Dengue Control Unit as of 15th of October [15–17].

Sri Lanka is a low-middle-income country. Its population is approximately 22 million with an annual per capita income of around USD 4000. In 2018, the yearly per capita healthcare expenditure was USD 157 [18]. The economic cost of dengue in Sri Lanka is complex. Reductions in labor efficiency resulting from prolonged illness and caregiving obligations lead to productivity losses, which in turn affect the economy as a whole [19]. Healthcare services, hospitalizations, and prescriptions are expensive, putting a strain on both individuals and the wider economy. Dengue prevention initiatives, such as vector control, public health campaigns, and other preventive interventions, all contribute to the disease's economic cost. Hospitals, particularly in high-burden areas like the Western Province, may struggle with resource allocations, leading to an overburdened healthcare infrastructure. Dengue outbreaks can strain public health infrastructure, leading to increased demand for healthcare services, hospital beds, and medical personnel. This strain impacts dengue patients, but it also affects the entire efficacy and effectiveness of the healthcare system, which jeopardizes the delivery of healthcare for other medical requirements [20]. The 2017 outbreak, with its high case count, resulted in significant economic losses affecting a variety of sectors. According to a 2020 study in Sri Lanka, the 20–29 age group had the

greatest dengue incidence rate, followed closely by the 10–19 age range, with no discernible gender difference. These age groups mostly comprise young working adults, students in higher education, and secondary school pupils. This age distribution is consistent with patterns reported in Southeast Asia, where older schoolchildren are at a higher risk of clinical disease [21,22]. Furthermore, according to previous literature, the majority of personnel-related costs recorded were reported as 79% for dengue control activities and 46% for hospitalizations out of all total costs. The per capita cost of the dengue control campaign was determined to be USD 0.42. The average cost of a hospitalization ranged from USD 216 to USD 609 for pediatric patients and from USD 196 to USD 866 for adult cases, depending on disease severity and treatment settings. These amounts focus on the financial allocation for controlling dengue and managing hospitalizations, offering useful insight into the economic elements of dengue-related expenditure in Sri Lanka [23].

*1.4. Unveiling Current Treatment Options and Vaccine Prospects for Dengue*

Dengue fever is a self-limited disease, and if it is not managed properly, it can even result in death through the development of DHF/DSS. Due to a lack of a suitable animal model that can outline the main features of human dengue disease, the pathogenesis of dengue infection remains poorly understood [24]. In the present situation, there is no specific antiviral therapy or treatment available for dengue fever. Several attempts have been made throughout the years to identify a specific antiviral medication for dengue infection. In such attempts, some investigational antiviral medications and techniques showed promise in preclinical investigations; however, their efficacy in human trials was limited. For example, a medicine known as chloroquine, which was previously used to treat malaria, showed antiviral activity against dengue in laboratory conditions but failed to show significant clinical advantages in human studies [25]. Further, balapiravir is an antiviral medication that was once thought to be a viable treatment for dengue. It acts as a nucleotide analogue, preventing dengue virus multiplication. Clinical trials were carried out to assess its efficacy, particularly in adult dengue patients; however, the results were not as encouraging as had been envisaged. Balapiravir did not fulfill the primary end point for efficacy in treating dengue in a randomized, double-blind, placebo-controlled trial [26]. As such, vigilant monitoring with supportive care is the principal course of treatment at present [27]. The clinical symptoms of dengue are usually managed by monitoring fluid balance and blood-clotting parameters with the administration of electrolyte supplements [28]. Studies have reported that when compared to the early infection phase, there is higher level of viremia in patients with severe dengue infection. This has led to the hypothesis that reducing viral load in blood via antiviral therapy in early stage of infection may reduce the severity of clinical conditions [29]. Despite the intensive effort of various researchers, to date there is no commercially available, effective therapy against all serotypes of dengue [30]. Efforts to develop antiviral drugs using phytochemicals and synthetic compounds are still ongoing [31].

At the present time, there is also a great deal of research on the development of a dengue vaccine. However, developing a successful vaccine for this disease has been extremely challenging. This is because a prospective vaccine needs to strike a balance between the virus's pathogenicity and the immunogenicity it elicits. In order to prevent dengue fever from getting worse after another infection, a vaccine must also be secure and offer total, long-lasting protection against all four serotypes. Up to now, there are five types of dengue vaccines that have been developed around the world: live attenuated vaccines, inactivated vaccines, recombinant subunit vaccines, viral-vectored vaccines, and DNA vaccines. Out of these, few vaccines and vaccine candidates have been developed against the virus that are currently used on humans. For instance, Dengvaxia, the first-ever vaccine against the dengue virus, was developed by the pharmaceutical company Sanofi Pasteur. It is a live attenuated vaccine that is recommended for children and adults aged 9 to 45 years who live in endemic areas and previously had dengue verified in a laboratory. Initially, it was approved for use in around 20 countries, and it was first licensed in Mexico in 2015.

However, lately it has been banned in some countries due to ADE [32]. The most recent registered dengue vaccine is QDENGA (TAK-003) by Takeda, a pharmaceutical company based in Japan. It is also a live attenuated vaccine that was first licensed in Mexico in 2022. This vaccine can be used on individuals between 4 and 60 years of age. To date, QDENGA remains safe and free from ADE [33]. The National Institute of Allergy and Infectious Diseases (NIAID) developed the TV003 and TV005 dengue vaccine candidates. These are also live attenuated, tetravalent vaccines, and they currently are in clinical trials [34]. There are a number of other dengue vaccine candidates in clinical development. They are DSV4, which was developed by the International Centre for Genetic Engineering and Biotechnology (ICGEB), India; and Butantan-DV, which was developed by the Butantan Institute [35–37].

While conventional medicines and vaccines may have limitations in efficiently combating dengue, a promising alternative lies in the realm of medicinal plants. Even though these natural remedies can effectively treat dengue symptoms, they may be utilized as an alternate treatment for dengue. Medicinal plants have the potential benefit of being widely available, as they are frequently locally sourced and easily cultivated. This is consistent with traditional medicine practices, which rely on readily available resources. Furthermore, sustainable medicinal plant gathering and cultivation can help to conserve biodiversity. However, to assure the dependability of these alternative therapeutics, difficulties such as the standardization of herbal formulations, quality control, and scientific validation of efficacy must be addressed [38]. The integration of traditional knowledge with modern scientific approaches, as well as community engagement, can enhance the accessibility and sustainability of using medicinal plants as an alternative therapy for dengue infection [39].

## 2. Methodology

A comprehensive search of the literature was conducted using the following databases and sources: Google Scholar, PubMed (US National Library of Medicine, Bethesda, MD, USA), Elsevier, ScienceDirect, Scopus, and Web of Science during the period of 15 January to 15 October 2023. The World Health Organization's official website was also used to gather a number of facts and figures. The following medical subject headings and keywords were included in the search: "Traditional medicine", "Sri Lanka", "Dengue", "Anti-dengue activity", "Anti-viral activity", "Herbal extracts", and "Biological compounds having anti-dengue activity". The following inclusion criteria were used: studies published in the English language (1997 to 2023) focusing mainly on traditional medicine practices and its potential anti-dengue activity, antiviral activity, microbial activity, clinical trials, case studies, and laboratory research. The following exclusion criteria were used: studies that were not related to traditional medicine, non-English publications, and publications older than 1997 or irrelevant to antiviral or microbial activity. The extraction of details was conducted using titles, author names, publication years, study designs, methodologies, anti-dengue and antiviral activities, and dosages and administrative methods of agents. This review discusses seven plant species used in Sri Lanka for the treatment of dengue.

### 2.1. Use of Medicinal Plants for the Treatment of Dengue Fever

Over 200 years ago, the isolation of morphine, the very first pharmacologically active compound, from the poppy plant (*Papaver somniferum*) initiated a whole new era wherein drugs can be purified from plants [40]. Nature has always been a rich source of ingredients that are beneficial in treating human health conditions [41]. Beginning with the earliest human civilizations, herbal medicinal formulae have been crucial as therapeutic agents. Compounds with medicinal value from natural products such as plants were isolated and characterized initially in the 19th century [42]. Despite neglecting them decades, scientists and researchers are now paying attention to natural products as potential drug candidates [43]. More than 80% of the population of the developing world depends on natural products as medicines because of their time-tested efficacy, safety, and cost-effectiveness. During the last 30 years, up to 50% of the approved drugs were derived

either directly or indirectly from natural products [28,42]. Over the past decade, plant-based medicines and related products have piqued the interest of the global market. Even though this interest and recognition can vary majorly from country to country, plant-based medicines are valued for their medicinal and economic benefits. Currently, international agencies and governments have directed their attention to investing in research based on traditional herbal medicine products [44].

The Sri Lankan traditional medicine practice "Deshiya Chikithsa/Hela Wedakama" dates back thousands of years. It has been playing a vital role in the cultural heritage of the country, providing healthcare to a major part of the population (to around 60–70% of the rural population) [45]. "Deshiya Chikithsa" is native to Sri Lanka and is the earliest system of medicine present in the country, even preceding the emergence of Ayurveda. According to the World Health Organization, there were around 20,353 registered Ayurvedic physicians and 8000 traditional practitioners in Sri Lanka in 2019. Most of the traditional medical practitioners are descendants of well-reputed family lines who engage in public healthcare with secret formulae and practices to cure diseases [44]. In traditional medicine, crude drugs are mostly parts of plants with medicinal values. The stem, root, wood, bark, flowers, leaves, fruits, seeds, and sometimes the whole plant is used to create medicinal formulae. Further, many plants are often used in combination with other plant sources as polyherbal formulae to treat various diseases on the island. Utilizing a blend of plants as opposed to single plants is more efficient and has a wider range of therapeutic applications [46]. There are nearly 30 different plant species reported in the literature that are claimed to have the potential to fight dengue and are used against the disease globally. Some of them are *Alternanthera philoxeroides*, *Carica papaya*, *Andrographis paniculata*, *Azadirachta indica*, *Cladosiphon okamuranu*, *Euphorbia hirta*, *Zingiber purpureum*, and *Momordica charantia*, where *Carica papaya* and *Euphorbia hirta* are the most commonly used plants throughout the world [47]. Active compounds found in the extracts of these plants, such as tannins, saponins, alkaloids, polysaccharides, phenols, and flavones, are reported to have antiviral activity [48].

This review focuses on a number of medicinal plants in Sri Lankan traditional medicine formulae with antiviral properties that are most often used to treat dengue viral fever in Sri Lanka.

### 2.2. Munronia pinnata

Kingdom—Plantae
Division—Magnoliophyta
Class—Magnoliopsida
Order—Sapindales
Family—Simaroubaceae
Genus—Munronia
Species—*Munronia pinnata* [49]

*Munronia pinnata* (Figure 1) is called "Bim-khohomba" in Sinhalese and "Nila-vempu" in Tamil [50]. In Sri Lanka, it is naturally found in dry, intermediate, and wet ecosystem zones. Due to overutilization of the wild population of the plant, it is currently considered an endangered plant in Sri Lanka [51]. It is 10–50 cm tall with a stem that is usually unbranched. This plant has a short stem and compound leaves that hover on a hard stem. Around 3, 5, 7, 9, or 11 leaflets can be found on a leaf. Its seeds are yellowish-gray in color [52].

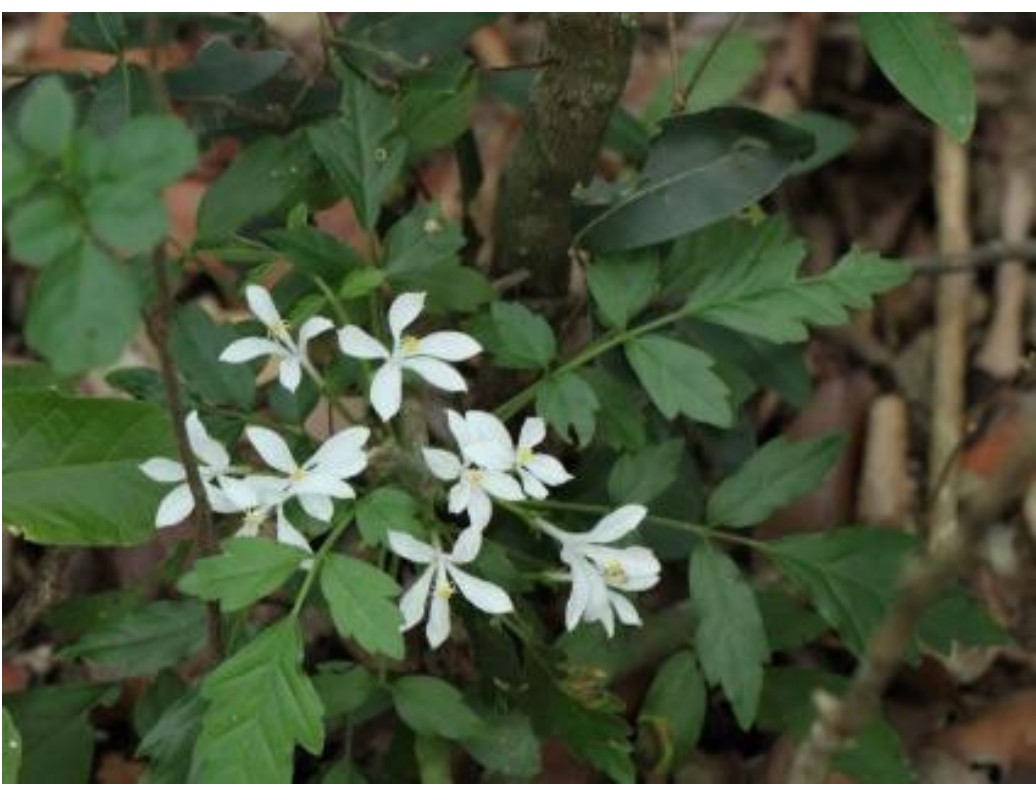

**Figure 1.** *Munronia pinnata* plant [53].

    A study was conducted in Sri Lanka in 2023 using the aqueous extract of the whole plant of *Munronia pinnata* to determine the anti-dengue activity. An MTT assay (3-(4,5-dimethylthiazol-2-yl)-2,5-diphenyltetrazolium bromide) was used to determine the maximum nontoxic dose (125 µg/mL) and 50% cytotoxic concentration ($CC_{50}$) (428.9 ± 21.55 µg/mL). The cell metabolic activity was measured using the MTT assay, which served as a gauge for cell viability, proliferation, and cytotoxicity. Further, a plaque-reduction test was also used to calculate the half-maximum inhibitory concentration ($IC_{50}$). This test can be used to test the activity of a potential antiviral treatment. The results of the above study revealed that this plant can inhibit DENV-1, DENV-2, and DENV-4 but not the DENV-3 serotype [54]. Furthermore, beta-caryophyllene, an active biological compound (Table 1) found in the plant, was reported to have a high binding affinity to dengue viral proteins during an in silico study conducted by Pajaro and colleagues [55].

**Table 1.** Chemical structures with antiviral activity in plants.

| Plant Name | Chemical Compound Present in the Plant | Structure of the Compound | Reference |
|:---:|:---:|:---:|:---:|
| *Munronia pinnata* | beta-Caryophyllene | | [56] |

**Table 1.** *Cont.*

| Plant Name | Chemical Compound Present in the Plant | Structure of the Compound | Reference |
|---|---|---|---|
| *Munronia pinnata* | Ganoderiol F | | [57] |
| *Azadirachta indica* | Gedunin | | [58] |
| | 8-(alpha,beta-Dimethylallyl) (Pongamol) | | [58] |
| *Cissampelos pareira* **L**. | Cycleanine | | [59] |
| | Laudanosine | | [59] |

**Table 1.** *Cont.*

| Plant Name | Chemical Compound Present in the Plant | Structure of the Compound | Reference |
|---|---|---|---|
| | Corytuberine |  | [59] |
| *Zingiber officinale* | Myrcenol |  | [59] |
| | 2-Heptanol |  | [59] |
| *Carica papaya* | 1′-OH-gamma-carotene glucoside/(Carotenoids B-G) |  | [59] |
| | Carpaine |  | [60] |
| *Curcuma longa* | Curcumin |  | [61] |
| | Myrtenol |  | [59] |

**Table 1.** *Cont.*

| Plant Name | Chemical Compound Present in the Plant | Structure of the Compound | Reference |
|---|---|---|---|
|  | (-)-beta-Curcumene |  | [59] |
| *Bambusa vulgaris* | Stigmasterol |  | [59] |
|  | Taxiphyllin |  | [59] |
|  | beta-Sitosterol |  | [59] |

Another study reported that beta-caryophyllene is a potent inhibitor of nonstructural protein1 (NS1) protein levels in the early stages of the infection, confirming that *Munronia pinnata* could inhibit the early stages of the DENV life cycle [62]. Moreover, according to prior literature, a triterpenoid named Ganoderiol F, an active biological compound (Table 1) in *Munronia pinnata*, has shown antiviral activity against the human immunodeficiency virus (HIV) through inhibiting HIV-1 protease [57]. This compound has shown similar activity when it was isolated from fruiting bodies of *Ganoderma lucidum* as well, and it has shown significant inhibitory activity against HIV-1 protease [63]. Recent research has demonstrated that beta-caryophyllene works as a naturally cannabinoid ligand. The cannabinoid system modulates numerous signaling pathways and mediates immune/inflammatory responses. Beta-caryophyllene produces therapeutic effects by activating cannabinoid type 2. Activated cannabinoids interact and crosstalk with peroxisome proliferator-activated receptors (PPARs). PPARs inhibit the replication of numerous viruses such as HIV, hepatitis B, hepatitis C viruses, and interestingly, the dengue virus as well. For instance, the PPARγ antagonist GW9662 has provided protection against dengue virus infection by inhibiting the viral load [64]. At the same time, PPARs have also been reported to reduce morbidity

and mortality in influenza A virus infections [56]. In addition to antiviral activity, this plant has been shown to have antibacterial and antifungal properties as well [65].

*2.3. Azadirachta indica*
Kingdom—Plantae
Division—Angiospermae
Class—Eudicots
Order—Sapindales
Family—Meliaceae
Genus—Azardirachta
Species—*Azadirachta indica* [49]

*Azadirachta indica* (Figure 2) is called as "Khohomba" in Sinhalese and "Vempu/Vembu/ Veppa" in Tamil [50]. *Azadirachta indica* can tolerate temperatures up to 50 °C, but it cannot tolerate the cold. Temperatures below 4 °C can kill the tree. It is well known for its ability to tolerate adversely dry environmental conditions, and the plant can survive in dry periods lasting seven to eight months. It can grow in infertile, rocky, and dry soils. However, salty or clayey and muddy soil are unsuitable for the plant. It can grow to a height of 15–20 m [66,67]. This plant is commonly found in many Sri Lankan household gardens. A computational experiment conducted in 2020 in India reported that small molecules of *Azadirachta indica* can potentially be useful against the dengue virus. The study analyzed the effect of small molecules of *Azadirachta indica* against the dengue virus by studying the molecular binding at the structural level. Gedunin and Pongamol (Table 1), two active biological compounds present in this plant, have shown strong interactions with important viral proteins. As such, these compounds are able to inhibit the receptor sites on the surface of human cells, where the DENV virus can be prevented from entering the body, and they leave no open receptor sites for DENV binding to occur [58]. Another study conducted in India reported that the crude aqueous extract of the leaf of the plant showed significant inhibitory activity against the dengue virus in vitro and in vivo; the maximum nontoxic concentration against C6/36 cells was 1.897 mg/mL, while for suckling mice, it was 120–30 mg/mL. A lack of clinical symptoms and a virus-specific amplicon have been observed in suckling mice pups that were infected with the dengue virus when they were inoculated with the leaf extract [68,69]. All the above studies have validated the efficacy of *Azadirachta indica* leaves in fighting the dengue virus. Further, the antiviral activities of aqueous extracts of *Azadirachta indica* have demonstrated effectiveness against a wide spectrum of viruses, such as the fowlpox, smallpox, polio, and herpes simplex viruses. Furthermore, the measles, Chikungunya, and vaccina viruses have also shown significant inhibition by aqueous extracts of *Azadirachta indica* [67]. A study conducted in 2016 using an ethanol extract of *Azadirachta indica* revealed that it had significant antiviral activity against the virus, which causes foot-and-mouth disease [70]. As a common practice in Sri Lanka, the leaves of this plant are used in several ways, such as during bathing when a person is infected with a viral infection such as chicken pox or measles, in almost every household.

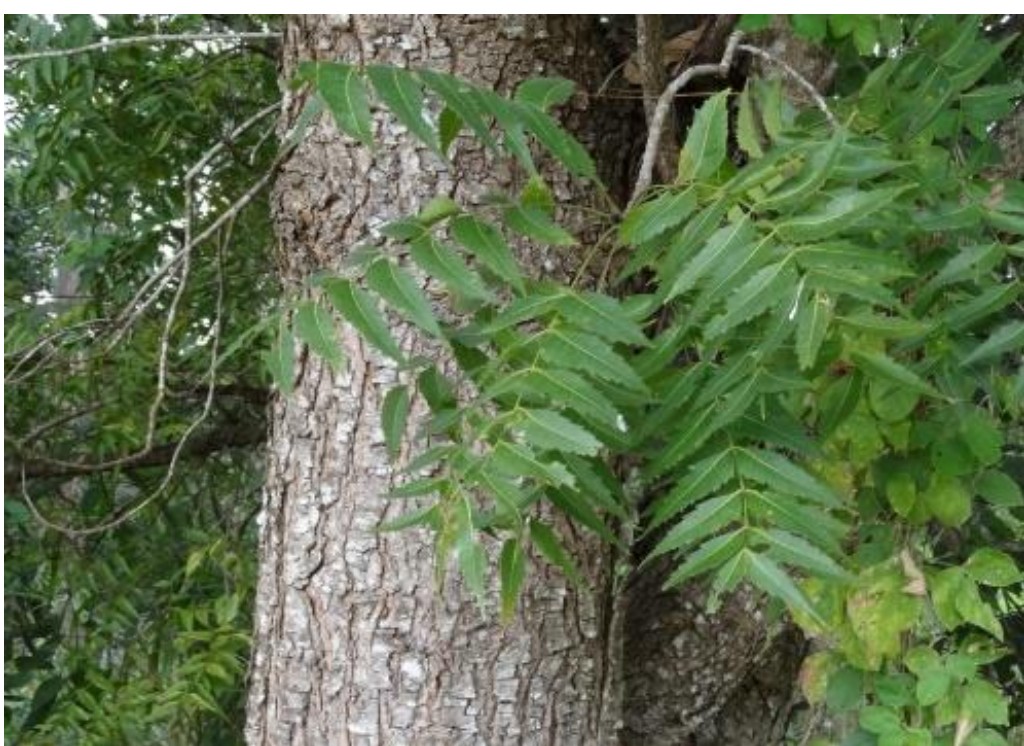

**Figure 2.** *Azadirachta indica* plant [53].

*2.4. Cissampelos pareira L.*
Kingdom—Plantae
Division—Magnoliophyta
Class—Magnoliopsida
Order—Ranunculales
Family—Menispermaceae
Genus—Cissampelos
Species—*Cissampelos pareira* [49]

 *Cissampelos pareira* L. (Figure 3), also known as "Diyamiththa" in Sinhalese and "Appatta/Ponmucuttai" in Tamil, is a widely distributed plant in Asia. It is a woody twiner with leaves that are 2.5 to 6 cm long and 2 to 7 cm wide [50,71]. A study reported inhibition of dengue virus replication in MCF-7 breast cancer cells by using the whole-plant extract of *Cissampelos pareira* L. The inhibition was shown to be dependent on estrogen receptor 1. Pyrimidine metabolism is notable among the pathways that this plant downregulates. Moreover, it has been noted that pyrimidine analogues inhibit a variety of viruses. As such, this plant extract may work by altering this route to prevent dengue virus infection in MCF-7 cells [72]. A further study conducted in 2015 by Sood and colleagues showed that a methanolic extract of the aerial part of this plant was able to inhibit all four DENV serotypes. The extract exhibited a dose-dependent inhibitory effect on NS1 antigen secretion, in which it was possible that the extract influenced the synthesis and release of NS1 antigen. The same study also proved that *Cissampelos pareira* L. extract exhibited a dose-dependent efficacy in the in vivo mouse model AG129, which is a promising dengue model. These mice are genetically engineered to lack functional interferon $\alpha/\beta$ and $\gamma$ receptors, which makes them highly susceptible to dengue virus infection. These researchers have patented the anti-dengue activity of this plant, and they have also developed a promising pharmaceutical candidate named "Cipa" that comprises *Cissampelos pareira* L. extracts [73,74]. Furthermore, a study has revealed that *Cissampelos pareira* L. has a high selective potential as an antibacterial, antifungal, and antituberculosis agent [75].

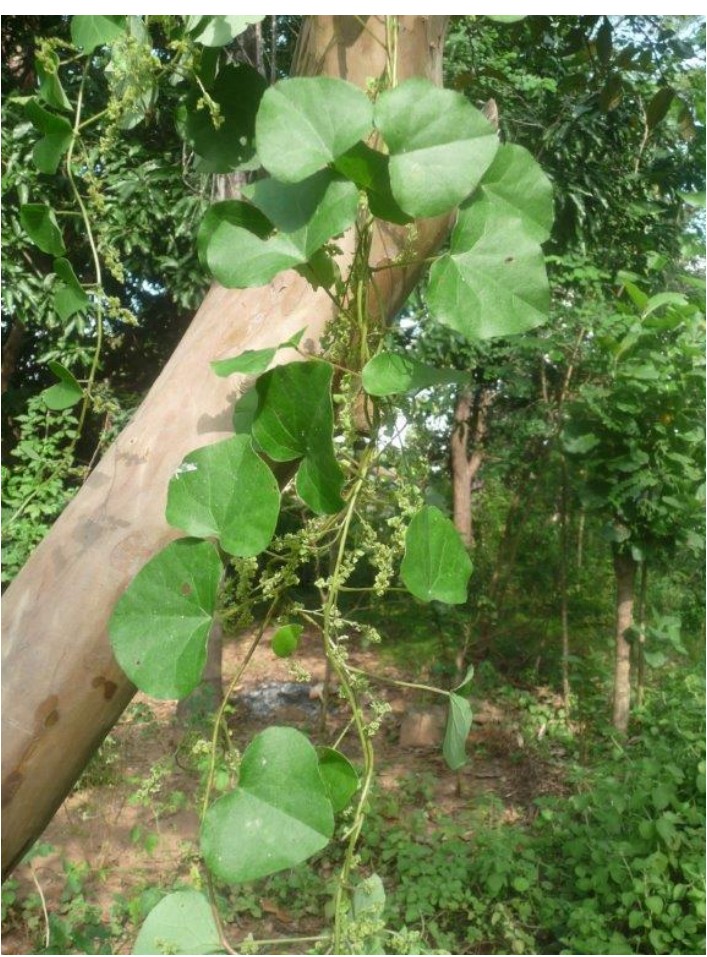

**Figure 3.** *Cissampelos pareira* L. plant [53].

*2.5. Carica papaya*

Kingdom—Plantae
Division—Angiospermae
Class—Eudicots
Order—Caricacea
Family—Caricaceae
Genus—Carica
Species—*Carica papaya* [49]

    *Carica papaya* (Figure 4), also known as "Gas-labu/Paepol" in Sinhalese and "Pappali/Pappayi" in Tamil, is a sparsely branched tree with spirally arranged leaves that may grow up to 5–10 m in height. This plant grows in tropical climates and is well known for its tasty fruit. It is very frequently found in many household gardens in Sri Lanka. *Carica papaya* leaves are known to have multiple chemical compounds that have anti-inflammatory and antioxidant properties. It is also believed to contain bioactive compounds that may help increase platelet production or prevent platelet destruction. Many research studies have been conducted with this plant and its anti-dengue activity [60]. A study conducted in 2016 by Zunjar and colleagues revealed that leaves of *Carica papaya* have many pharmacological and therapeutic potentials in the treatment of dengue fever. According to the researchers, *Carica papaya* leaf extract has immunomodulatory activities and can elevate the platelet count in dengue-infected patients. This was due to the antithrombocytopenic efficacy of the *Carica papaya* L. leaf alkaloid extract and the bioactive component carpaine (Table 1) that was extracted from the leaves. They further reported that *Carica papaya* leaf extract has the ability to reduce the production and expression of various cytokines such as IL-2, IL-4, IL-5, eotaxin, NF-κB, iNOS, and TNF-α. These factors are vital to the immune system

and are crucial in coordinating the body's defense against pathogens. Maintaining good health requires these components to be used in a balanced and coordinated manner [60,76]. A study conducted in Sri Lanka in 2012 tested the in vitro erythrocyte membrane stabilization capability of *Carica papaya* leaf extract on dengue-infected individuals' blood. The researchers reported that a freshly prepared extract (37.5 μg/mL) significantly inhibited the hemolysis of red cells, suggesting a potential therapeutic effect on disease processes that cause destabilization of biological membranes, such as dengue [77]. There are also a number of studies conducted with humans. Venugopal (2018) found that patients treated with papaya leaf extract (1100 mg three times daily for five days) had an early increase in platelet count and a shorter average length of hospital stay. Patients treated with the extract had an average hospital stay of 5.42 days, while patients in the control group had an average hospital stay of 7.2 days. Additionally, the control group required more platelets from blood transfusions than the treated group. Furthermore, another study carried out in 2019 on dengue-infected children and teenagers aged 1 to 16 years reported that administration of *Carica papaya* leaf extracts showed an increasing trend in the platelet count. *Carica papaya* leaf extract was given three times a day for five days in the proper dosage forms of tablets (1100 mg) for children over 12 years old, syrup (275 mg/5 mL) at 10 mL for children between 6 to 12 years and 5 mL for children under 6 years old. The change was faster in the test group compared to the control group, who only received the supportive care given to dengue patients [78]. Moreover, a study conducted in India reported that *Carica papaya* leaf extracts significantly increased the platelet count in dengue-infected individuals without any side effects. They used a capsule made from *Carica papaya* leaf extracts in dengue patients at 500 mg once daily with supportive care for consecutive 5 days [79].

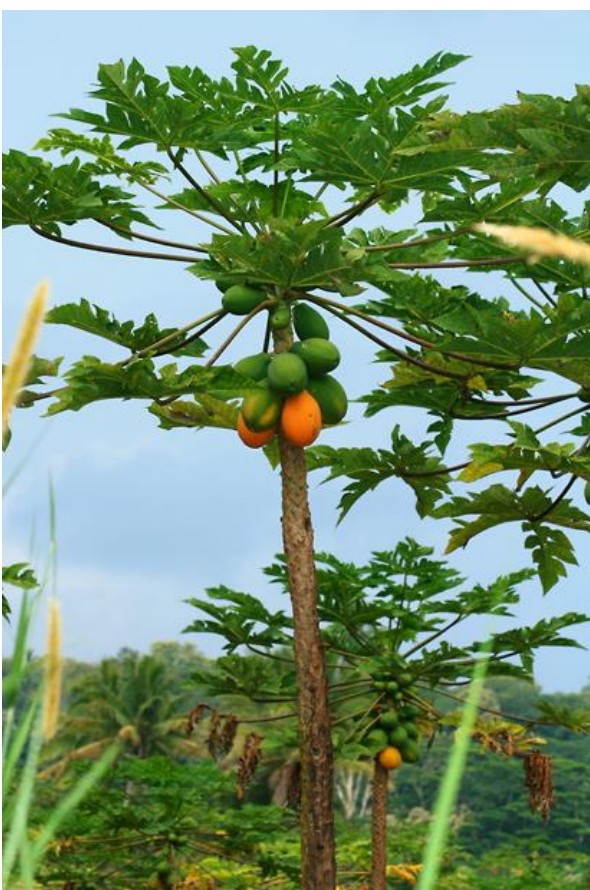

**Figure 4.** *Carica papaya* plant [53].

*2.6. Zingiber officinale*

Kingdom—Plantae
Division—Angiospermae
Class—Monocots
Order—Zingiberales
Family—Zingiberaceae
Genus—Zingiber
Species—*Zingiber officinale* [49]

*Zingiber officinale* (Figure 5), which is believed to be native to India, is commercially and widely grown in South and Southeast Asia, Africa, the Caribbean, Latin America, and Australia [80,81]. It is called "Inguru" in Sinhalese and "Injzi/Chukku" in Tamil. It is widely grown throughout much of Sri Lanka. The rhizome is mainly used in traditional medicine and is also used for day-to-day culinary purposes in the country. A study conducted in 2015 demonstrated that an aqueous extract of *Zingiber officinale* rhizomes regulated plasma leakage in dengue virus infection by inhibiting the expression and activity of the matrix metalloproteinases MMP-2 and MMP-9, which play key roles in promoting vascular permeability leading to hypovolemic shock in DHF and DSS. At the same time, a *Zingiber officinale* extract upregulated the expression of the metalloproteinase tissue inhibitors TIMP-1 and TIMP-2, which regulate matrix metaloproteinases. The above study also reported a strong dose-dependent modulatory activity of an aqueous extract of *Zingiber officinale* on the gene expression of TIMP-1 and TIMP-2 metaloproteinases [82]. Another study conducted in 2020 illustrated excellent anti-Chikungunya activity. This study was conducted using an aqueous extract of *Zingiber officinale* rhizomes on cell cultures infected with the Chikungunya virus [83].

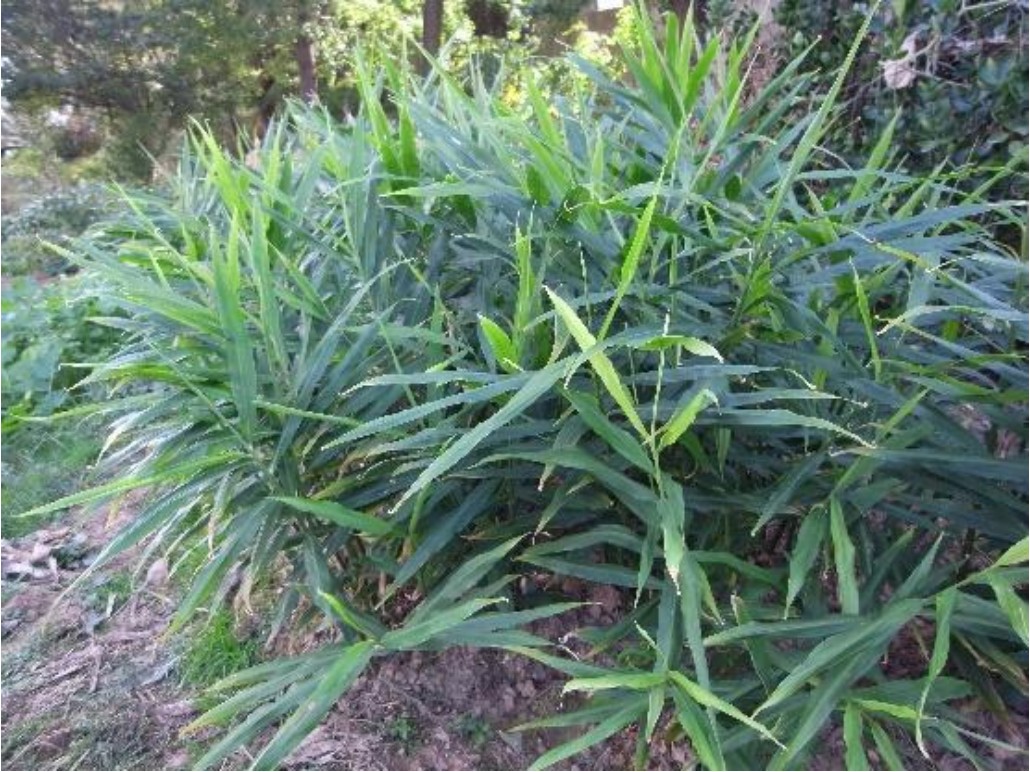

**Figure 5.** *Zingiber officinale* plant [53].

*2.7. Curcuma longa*

Kingdom—Plantae
Division—Angiospermae
Class—Monocots
Order—Zingiberales
Family—Zingiberaceae
Genus—Curcuma
Species—*Curcuma longa* [49]

    *Curcuma longa* (Figure 6) is called "Kaha" in Sinhalese and "Mancal/Manjal" in Tamil. Its rhizome is used for traditional medicinal purposes and culinary purposes in Sri Lanka and in most South Asian countries. The main active compound in *Curcuma longa* is curcumin (Table 1), which is reported to have outstanding antiviral activity against many viruses, such as the dengue virus, Epstein–Barr virus, and HIV [84,85]. Antiviral activity against the influenza virus ($H_1N_1$ and $H_6N_1$) and the herpes simplex virus types 1 and 2 (HSV-1 and -2) has also been reported [61]. A study conducted in 2017 reported results for both in vitro and in vivo experiments. They found an $IC_{50}$ of 17.91 µg/mL for the *Curcuma longa* extract, while they found a $CC_{50}$ value of 85.4 µg/mL against Huh-7 it-1 cells (a hepatocellular carcinoma cell line), with a selectivity index of 4.8 showing dengue viral inhibition at high concentrations of the extract. Further, an extract of *Curcuma longa* at a dose of 0.147 mg/mL showed antiviral effects against DENV-2 and a reduced viremia period in mice in the in vivo study [86]. Furthermore, another study reported that curcumin inhibited the production of viral particles in a dose-dependent manner in BHK-21 cells, which are derived from hamster kidneys infected with the DENV-2 serotype, with a selective index of 2.56. The authors suggested that this inhibition may not be due to a direct effect on viral production but rather to the effects of curcumin on various cellular systems such as the ubiquitin–proteasome system or the cytoskeleton's actin filaments, leading to apoptosis or programmed cell death [84]. Further, EGYVIR, an immunomodulatory herbal extract created by a group of scientists from Egypt using *Curcuma longa,* proved to have potent antiviral activity against the severe acute respiratory syndrome coronavirus 2 (SARS-CoV-2) in research conducted in 2020 using Huh-7 cells [87].

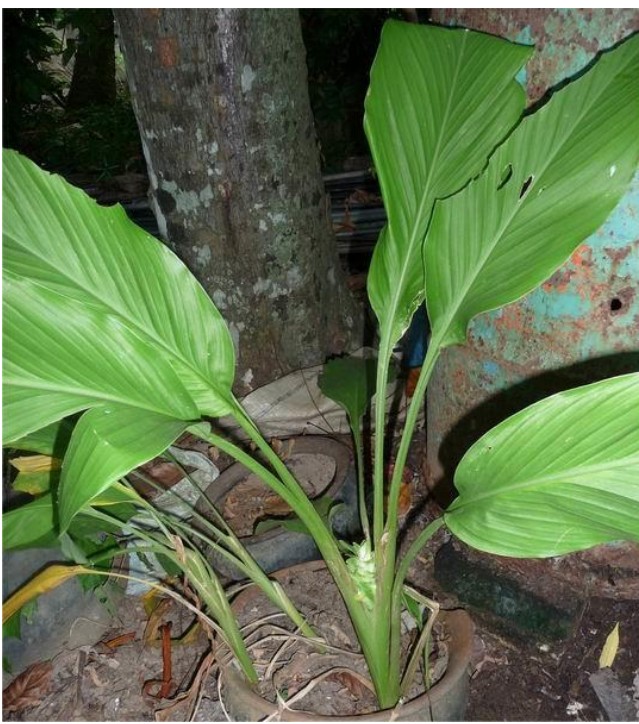

**Figure 6.** *Curcuma longa* plant [53].

*2.8. Bambusa vulgaris*

Kingdom—Plantae
Division—Angiospermae
Class—Monocots
Order—Poales
Family—Poaceae
Genus—Bambusa
Species—*Bambusa vulgaris* [49]

    *Bambusa vulgaris* (Figure 7) is called "Una" in Sinhalese and "Moongil" in Tamil [88]. Bamboo is classified according to the species, type, and variety. Bamboo is basically divided into two groups: herbaceous and woody bamboo. *Bambusa vulgaris* is a woody bamboo. The fastest-growing plant in the world is also the bamboo tree [89]. At present, this plant is the most widely cultivated bamboo type throughout the subtropics and tropics. It grows well in areas with permanent humid conditions, like riverbanks and lakesides. A study conducted in Sri Lanka on patients infected with the dengue virus revealed that bamboo extract effectively controlled signs and symptoms in the patients when they were administered one dose per day regardless of the length and severity of the disease. Platelet count elevation and an increase in the hematocrit count have been observed within 48 h after starting the treatment [90]. Another experiment conducted in 2009 in Nigeria revealed that the measles virus was susceptible to the ethanol extract of *Bambusa vulgaris* in vitro, but the same extract was not effective against the yellow fever virus and poliovirus type 1 [91]. Stigmaterol (Table 1) is an active biological compound present in bamboo. A study conducted in 2014 by Erine and colleagues reported that two synthetic stigmasterol derivatives were found to exhibit significant antiviral activity against HSV-1 replication and spreading in human epithelial cells derived from ocular tissues. In all cases, both compounds prevented HSV-1 multiplication when added after infection and virus propagation [92].

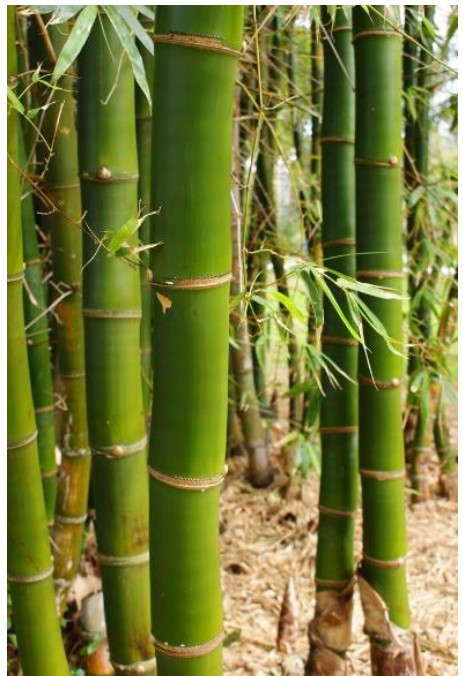 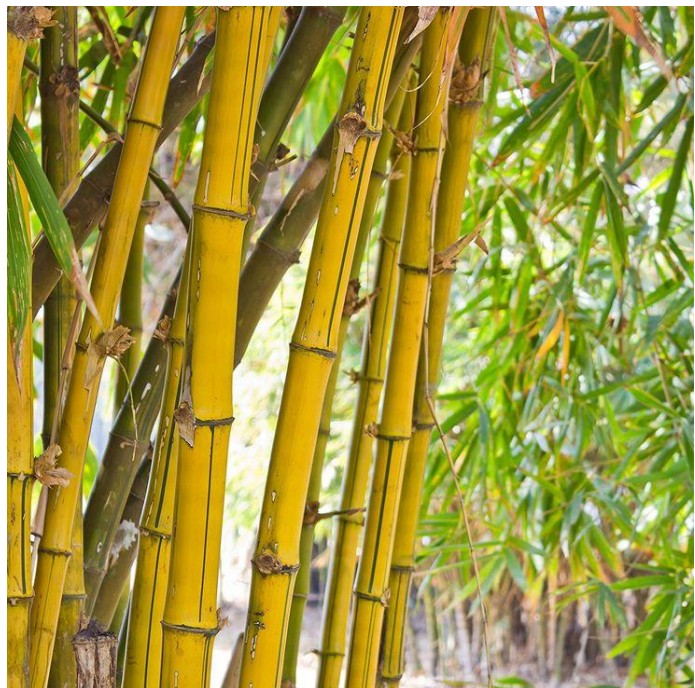

**Figure 7.** *Bambusa vulgaris* plant [53].

## 3. Conclusions and Recommendations

    This review sheds light on the rich traditional knowledge of medicinal plants in Sri Lanka; specifically, their relevance in the context of managing dengue fever. The findings highlight the usage of seven plants in local traditional medicine practices, showing the

intimate connection that communities have with their natural environment. The use of medicinal plants is firmly rooted in traditional healthcare systems around the world, providing a varied array of therapeutic chemicals. However, geographic, cultural, and ecological factors influence the accessibility and sustainability of these therapies [93,94]. Thus, this review emphasizes the need for sustainable practices in harvesting and utilizing medicinal plants to ensure the preservation of biodiversity and the long-term availability of these resources. As medicinal plant remedies gain acceptance, concerns regarding their competitiveness with pharmaceutical agents also arise. The pharmaceutical industry's commitment to rigorous testing, standardization, and regulatory approval highlights conventional medications' efficacy and safety. Medicinal plants, on the other hand, may face difficulties in meeting these demanding criteria [93,95,96]. Nevertheless, recognizing the importance of medicinal plants in traditional medicine in the overall healthcare landscape can contribute to the development of complementary and integrative approaches for managing dengue fever and other health challenges, which may help to reduce the overload in a clinical setting.

In traditional Sri Lankan medicinal practice, medicinal formulae contain more than one medicinal plant in their content. This polyherbal formulae will act in a synergetic manner that increases the effectiveness of the therapies while at the same time diminishes their side effects. Some of these formulations have been practiced for years, and they provide excellent information regarding potential safe and effective drugs that can be used in the clinic. Nevertheless, the promising aspects of certain medicinal plants in combating dengue fever symptoms lie in their demonstrated ability to alleviate fever, reduce inflammation, decrease the viral load, increase platelet counts, and enhance the overall immune response in a patient. A comprehensive amount of research has been conducted on plants and their antiviral activities, but it is crucial to approach these findings with a balanced perspective. As such, further research is warranted to scientifically validate the efficacy and safety of these traditional remedies to establish a bridge between traditional knowledge and modern medicine. Collaborative efforts between traditional healers, scientists, and healthcare professionals can facilitate the integration of valuable insights from traditional medicine into evidence-based healthcare practices. In the broader context, this review contributes to the growing body of knowledge that explores the potential of medicinal plants as complementary or alternative treatments for viral diseases. It encourages a holistic perspective that values both traditional wisdom and scientific rigor, fostering a collaborative approach for the benefit of public health.

**Author Contributions:** Conceptualization, J.K.H. and A.I.K.; methodology, J.K.H.; software, J.K.H.; validation, J.K.H.; resources, J.K.H.; data curation, J.K.H.; writing—original draft preparation, J.K.H.; writing—review and editing, J.K.H. and A.I.K.; visualization, N.K., C.L.G. and A.I.K.; project administration, A.I.K.; funding acquisition, A.I.K. All authors have read and agreed to the published version of the manuscript.

**Funding:** We would like to thank General Sir John Kotelawala Defence University, Ratmalana, Sri Lanka (grant number KDU/RG/2021/CARE/008) for the funding that was provided to AIK to fund a MPhil research project, and The APC was also funded by General Sir John Kotelawala Defence University, Ratmalana, Sri Lanka.

**Institutional Review Board Statement:** Not applicable.

**Informed Consent Statement:** Not applicable.

**Data Availability Statement:** Not applicable.

**Conflicts of Interest:** The authors declare no conflicts of interest.

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
