# Peer review of "Medicinal Plants Used in Sri Lankan Traditional Medicine for Dengue Fever"

_2036-7481, doi:10.3390/microbiolres15020032_

Round 1

Reviewer 1 Report

Comments and Suggestions for Authors

This article provides a "systematic literature review" on an interesting topic in public health. This was left to add in the title, in addition to the fact that a total of 6 plant species were considered.

Some other details:

- on line 132: date or data extraction?

- taxonomy: add the families of the plants considered;

- line 287: number? Which?

- rewrite lines 299-300;

- line 313: children and teenagers;

- line 362: add a point;

- lines 398-401: topic not mentioned before in the text.

Author Response

Dear Reviwer,

On behalf of my co-authors, I would like to thank you very much for sending us the reviewer comments to correct our manuscript (microbiolres-2763408 Type of manuscript: Review, Title: Medicinal plants used in Sri Lankan traditional medicine for dengue fever) further, to be published in your journal. We have revised our manuscript according to your‘comments. 

I warrant that this manuscript has been read by all co-authors and I have submitted this manuscript on behalf of all the authors with their full consent.  The authors declare no conflicts of interest.  I would be very appreciated if you could consider accepting our manuscript for publication in your journal.

Thank you for your consideration.

With kind regards,

Dr. Anchala I. Kuruppu (PhD) on behalf of all co-authors.

Principle Investigator/Senior Lecturer.

Reviewer 

on line 132: date or data extraction? - Corrected

taxonomy: add the families of the plants considered- Added

line 287: Number of research studies have been conducted with this plant and its anti-dengue activity.”- Removed the word Number.

rewrite lines 299-300 -  Corrected and rewritten.

line 313: children and teenagers - Corrected

line 362: add a point; - Added

lines 398-401: topic not mentioned before in the text- This is now clearly added to the introduction section.

Reviewer 2 Report

Comments and Suggestions for Authors

The manuscript titled "Medicinal plants used in Sri Lankan traditional medicine for dengue fever" reviews literature on the use of medicinal plants to treat dengue in Sri Lanka. The authors provide information to support the use of medicinal plants in the treatment plan for Dengue cases. This could be a useful tool in Sri Lanka since effective, licensed Dengue therapies and vaccines are not yet available. The review of various plants is thorough and accessible to the general public. And images of the plants are appreciated. However, the introduction and discussion sections need to be expanded to support the use of medicinal plants to treat Dengue. I have provided the following comments for consideration.

Line 12: Is the word Dengue to be bolded?

Section 1.2- Expand on the discussion of Dengue burden. What are the financial/economical impacts? What about health care system impacts? The authors provide figures of cases and deaths during recent outbreaks. How do the size of the outbreaks compare to other countries that are impacted by this virus? Please expand on this section to emphasize the true burden of this virus in Sri Lanka.

Section 1.4 Authors state there are no viral therapies or treatments. Please expand on treatments and therapies that have been tried and explain why they failed. This will then help support the use of medicinal plants as a future therapy and drug development. 

Also, please add a transition paragraph after the discussion of vaccines. The manuscript jumps right into the methods and provides no context on the use of medicinal plants for Dengue therapy.

Conclusion and Recommendations:

Please add a  more in-depth discussion on accessibility and sustainability of using medicinal plants as an alternative therapy. There are several treatments and vaccines in research. If these products do make it to market, how well will medicinal treatment compete with pharmaceutical products? How can they enhance current dengue therapy research? Please discuss topics such as cost, access, production/manufacturing and distribution of medicinal plant therapy to the general population. 

Author Response

Dear Respected Reviwer, 

On behalf of my co-authors, I would like to thank you very much for sending us the reviewer comments to correct our manuscript (microbiolres-2763408 Type of manuscript: Review, Title: Medicinal plants used in Sri Lankan traditional medicine for dengue fever) further, to be published in your journal. We have revised our manuscript according to your valuable ‘comments.

I warrant that this manuscript has been read by all co-authors and I have submitted this manuscript on behalf of all the authors with their full consent.  The authors declare no conflicts of interest.  I would be very appreciated if you could consider accepting our manuscript for publication in your journal.

Thank you for your consideration.

With kind regards,

Dr. Anchala I. Kuruppu (PhD) on behalf of all co-authors.

Principle Investigator/Senior Lecturer.

Line 12: Is the word Dengue to be bolded? Corrected and removed bold.

Section 1.2- Expand on the discussion of Dengue burden. What are the financial/economical impacts? What about health care system impacts? The authors provide figures of cases and deaths during recent outbreaks. How do the size of the outbreaks compare to other countries that are impacted by this virus? Please expand on this section to emphasize the true burden of this virus in Sri Lanka - Corrected and we have added more text to section 1.2

Section 1.4 Authors state there are no viral therapies or treatments. Please expand on treatments and therapies that have been tried and explain why they failed. This will then help support the use of medicinal plants as a future therapy and drug development-  Corrected and added more text to section 1.4 as suggested

Also, please add a transition paragraph after the discussion of vaccines. The manuscript jumps right into the methods and provides no context on the use of medicinal plants for Dengue therapy. – Corrected as suggested

Conclusion and Recommendations:

Please add a more in-depth discussion on accessibility and sustainability of using medicinal plants as an alternative therapy. There are several treatments and vaccines in research. If these products do make it to market, how well will medicinal treatment compete with pharmaceutical products? How can they enhance current dengue therapy research? Please discuss topics such as cost, access, production/manufacturing and distribution of medicinal plant therapy to the general population- Corrected as suggested the conclusion and recommendations.

Reviewer 3 Report

Comments and Suggestions for Authors

Dear authors,

Your review presents an overview of dengue, its clinical presentations, the dengue burden in Sri Lanka and the potential of Sri Lankan medicinal plants used in traditional medicine for treating dengue.

It is enough well written, but I have some remarks:

General – the style of references should be brought to a single standard throughout the text.

Lane 32 - “These two types of mosquitoes” – not “types” but “species”. Correct, please.

Lane 33 - “Both of these species have white color strands on their legs.” - What is the message in this information? I think this can be removed.

Section 1.2 - You do not mention the main danger of getting dengue fever again, especially if it is caused by a different serotype of the virus. Please add this information, it would be very useful for the readers.

Lane 217 and onwards – “I. Azadirachta indica” – What does ”I.” mean here?

References section – some references contains no information where it could be accessed (for example, reference 113 and so on). Provide the correct references, please. Also, it seems that reference to Wikipedia is not acceptable

Comments on the Quality of English Language

Some correction is needed.

Author Response

Dear Respected Reviwer

On behalf of my co-authors, I would like to thank you very much for sending us the reviewer comments to correct our manuscript (microbiolres-2763408 Type of manuscript: Review, Title: Medicinal plants used in Sri Lankan traditional medicine for dengue fever) further, to be published in your journal. We have revised our manuscript according to your ‘comments.

I warrant that this manuscript has been read by all co-authors and I have submitted this manuscript on behalf of all the authors with their full consent.  The authors declare no conflicts of interest.  I would be very appreciated if you could consider accepting our manuscript for publication in your journal.

Thank you for your consideration.

With kind regards,

Dr. Anchala I. Kuruppu (PhD) on behalf of all co-authors.

Principle Investigator/Senior Lecturer.

General – the style of references should be brought to a single standard throughout the text. Corrected

Lane 32 - “These two types of mosquitoes” – not “types” but “species”. Correct, please- Corrected

Lane 33 - “Both of these species have white color strands on their legs.” - What is the message in this information? I think this can be removed- This was removed as suggested.

Section 1.2 - You do not mention the main danger of getting dengue fever again, especially if it is caused by a different serotype of the virus. Please add this information, it would be very useful for the readers. – Corrected as suggested and we have added about ADE

 Lane 217 and onwards – “I. Azadirachta indica” – What does ”I.” mean here?- Corrected

References section – some references contain no information where it could be accessed (for example, reference 113 and so on). Provide the correct references, please. Also, it seems that reference to Wikipedia is not acceptable- Corrected